



# The Earth Science Box Modeling Toolkit (ESBMTK)

Ulrich G. Wortmann[1], Tina Tsan[1], Mahrukh Niazi[2], Ruben Navasardyan[1], Magnus Roland Marun[1],
Bernardo S. Chede[4], Jingwen Zhong[1], and Morgan Wolfe[3]

[1]University of Toronto, Canada
[2]Orbio Earth, Cologne, Germany
[3]University of Southampton, United Kingdom
[4]Federal Fluminense University, Brazil

**Correspondence:** Ulrich G. Wortmann (uli.wortmann@utoronto.ca)

**Abstract.** The Earth Science Box Modeling Toolkit (ESBMTK) is a Python library designed for building and analyzing box models in Earth science. It uses a modular, object-oriented approach to study topics like the long-term carbon cycle and the impact of atmospheric $CO_2$ changes on seawater chemistry. ESBMTK allows users to define models in a straightforward and readable way, which also serves as documentation. These model definitions are then converted into equations and solved using

standard numerical libraries. The toolkit includes features for common box modeling tasks such as gas exchange between the ocean and atmosphere, marine carbonate chemistry, and isotope calculations. ESBMTK has been effectively used in both teaching and research settings. While the library is continually being improved, its core interface is stable and comes with extensive documentation.

## 1 Introduction

Box modeling is a versatile tool to explore a variety of earth systems processes. Their modest hardware requirements facilitate their use for teaching, or to investigate problems that require long integration times. Prominent examples include e.g., the Harvardton-Bear type models to explore aspects of the marine carbonate system (e.g., Broecker et al., 1999), the GEO-CARBSULF model which describes the evolution of the carbon, oxygen, and sulfur biogeochemical cycles over Phanerozoic times Berner (2006), or the LOSCAR model, which models the atmospheric and marine carbon system components and their

C-isotope ratios Zeebe (2012). Even limiting the citations to a specific subject area like paleoceanography, results in a long list of publications demonstrating the importance of box modeling, (e.g., Sarmiento and Toggweiler 1984; Tyrrell 1999; Wallmann 2003; Ridgwell 2003; Tyrrell and Zeebe 2004; Archer 2005; Wortmann and Chernyavsky 2007; Slingerland and Kump 2011; Markovic et al. 2015; Bachan and Kump 2015; Luo et al. 2016; Rennie et al. 2018; Yao et al. 2018; Boudreau et al. 2018, 2019; Shields and Mills 2021; Mills et al. 2021; Paytan et al. 2021; Shields and Mills 2021)

Box models, unlike more complex earth system models, require minimal computational resources. This allows researchers to focus on specific aspects of the earth system, e.g., how carbonate sediment dissolution mitigates ocean acidification. However, many undergraduate and graduate earth science students lack proficiency in traditional coding languages and differential equation solving, which can limit the use of box models in classroom settings. However, the simplicity and widespread adoption





of Python, along with the availability of cloud-based computing environments like Jupyter Notebooks, have expanded coding

accessibility beyond traditional audiences. Here, we introduce a Python library, that separates model geometry (and processes) from the underlying numerical implementation, and thus allows students (and researchers) to focus on the conceptual challenges, rather than mathematical theory. We successfully used this library in undergraduate and graduate teaching, as well as for ongoing research projects.

Our approach is best demonstrated by a simple example. Box models are formulated as a system of coupled ordinary

differential equations (ODE), that describe e.g., the transfer of matter between reservoirs (boxes). To give a trivial example (following Glover et al. 2011), let's consider the concentration of phosphate in a two-box ocean. The concentration change of phosphate in the surface box is simply a function of the phosphate fluxes into and out of the box :

$$\frac{d[PO_4]_S}{dt} = \frac{F_w + F_u - F_d - F_{POP}}{V_S} \tag{1}$$

where $F_w$ denotes the $PO_4$ weathering flux, $F_u$ the $PO_4$ upwelling flux, $F_d$ the $PO_4$ flux related to the thermohaline circulation,

$F_{POP}$ the $PO_4$ uptake by primary production, and $V_S$ denotes the volume of the surface box.

While conceptually simple, translating the above into computer code is often beyond the coding skills of many earth science students. Furthermore, with increasing model complexity, the reverse process, i.e., deriving the governing relationships from the program code, becomes considerably more difficult. The Earth Science Box Modeling Toolkit (ESBMTK) aims to address both problems by facilitating a declarative model definition that also serves as the model documentation. Modeling objects

(instances in Python) are created by importing the respective ESBMTK classes which are then used to create e.g., reservoir objects. Listing 1 shows how to import the classes, create reservoirs and define their relationships.

Class instances can then be combined to build a model, e.g., a reservoir instance (say for the surface ocean box), which can be connected to a second reservoir instance (e.g., the atmosphere box) via a connection instance that specifies their relationship (e.g., gas exchange). This results in a hierarchical structure, that, while verbose, explicitly encodes the model geometry and the

relationships between the respective model objects (see Fig. 1).

ESBMTK comes with a wide array of predefined processes to connect boxes (e.g., scale a flux relative to another flux, isotope effects, sediment dissolution etc.). Additionally ESBMTK provides a variety of methods for post-processing, and data management (including graphical output), and leverages standard Python methods for introspection and interactive documentation (see the user guide for details https://esbmtk.readthedocs.io/) While there is no graphical interface similar to Simulink, this

approach significantly reduces coding complexity and model development time. Crucially, the model structure is independent of the numerical implementation. Instead, the model is parsed dynamically to create the necessary equation system which is then passed to an ODE solver library like ODEPACK. Separating model description from numerical implementation results in well-documented model code, and combines the computational efficiency of state-of-the-art numerical libraries with the ease of use of Python. Presently, the resulting ODE is coded as Python, but it is possible to modify the parser to output the ODE

system in other languages (e.g. Julia).





---

**Listing 1** Code fragment showing how to import ESBMTK classes and create Reservoir objects (instances). The `ConnectionProperties` instance defines the relationship between Reservoirs. In this case, a flux that depends on the volume of water and concentration of a given species in the upstream reservoir. Note that the name of the `ConnectionProperties` instance is set implicitly.

---

```python
from esbmtk import Model, Reservoir, ConnectionProperties

M = Model(
    stop="3 Myr",  # end time of model
    max_timestep="1 kyr",  # upper limit of time step
    element=["Phosphor"],  # list of element definitions
)
Reservoir(
    name="S_b",  # box name
    species=M.PO4,  # species
    register=M,  # this box will be available as M.S_b
    volume="3E16 m**3",  # surface box volume
    concentration="0 umol/l",  # initial concentration
)
Reservoir(
    name="D_b",
    species=M.PO4,
    register=M,
    volume="100E16 m**3",
    concentration="0 umol/l",
)
ConnectionProperties(
    source=M.D_b,  # source of flux
    sink=M.S_b,  # target of flux
    rate="30 Sv",  # rate of flux
    ctype="scale_with_concentration",
    id="Thermohaline",  # connection id
)
```

---



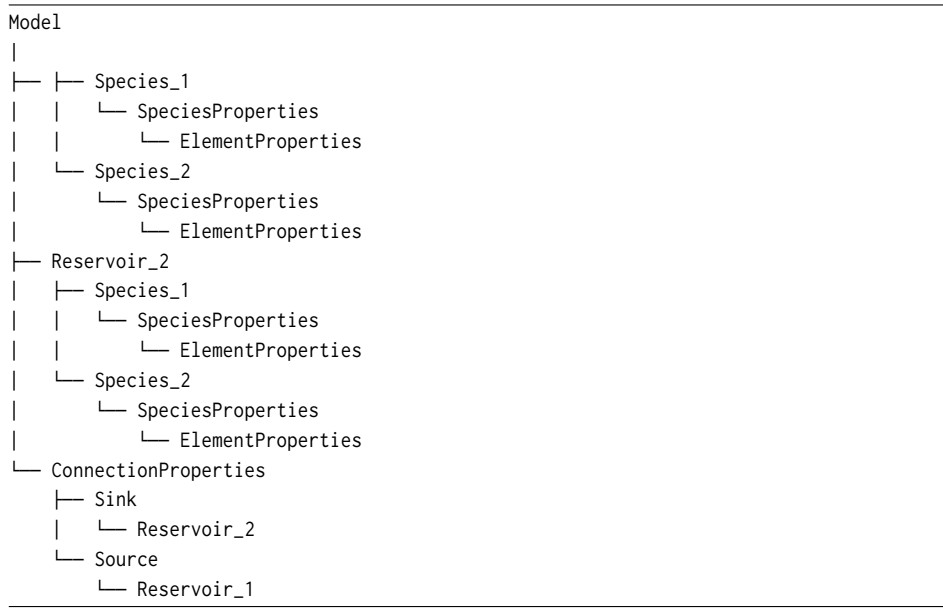

**Figure 1.** ESBMTK uses a modular hierarchical object structure. Changing, e.g., the type and number of species in the boxes, will therefore automatically propagate to the `ConnectionProperties` instance (unless it specifically limits to which species it apply). caption

## 2 Methods

The following sections are not meant as a user guide, rather, they describe implementation details and the underlying assumptions. The user guide and code examples are available online, see the Code Availability section below.

### 2.1 Isotope ratios

Several ESBMTK classes have the option to perform stable-isotope-related calculations. This requires that the respective element definitions contain the necessary data, e.g., the isotope ratios of the respective standards, labels, and the name of the isotope scale. For details, see the `species_definitions.py` and the user guide. In the following, I will only describe the pertinent implementation details.

To specify the initial isotope ratio of a given reservoir instance, ESBMTK uses the common delta notation, e.g., for sulfur
$^{34}$S and $^{32}$S we can write:

$$\delta^{34}S = \left( \frac{\left( \frac{^{34}S}{^{32}S} \right)_{\text{Sample}}}{\left( \frac{^{34}S}{^{32}S} \right)_{\text{Standard}}} - 1 \right) \times 1000 \quad [\text{mUr} \quad \text{VCDT}] \tag{2}$$

The unit is in permil (i.e., per thousand) or milli Urey where 1‰ = 1 mUr (Brand and Coplen, 2012). It is customary to combine the unit with the name of the reference standard (e.g., [mUr VCDT]), however, ESBMTK currently does not parse isotope units, rather, delta values a simply given a numeric value where 1 equals 1 mUr.





If a connection between two reservoirs involves a process that changes the isotope ratio, one can specify the enrichment factor $\epsilon$ in the `ConnectionProperties`, where $\epsilon$ is defined as

$$\epsilon = (\alpha - 1) \cdot 1000 \quad [\text{mUr}] \tag{3}$$

where $\alpha$ equals the isotope fractionation factor between two substances like HCO$_3^-$ and organic matter (OM) during photosynthesis,

$$75 \quad \alpha_{HCO_3^- - OM} = \frac{\left(\frac{^{13}C}{^{12}C}\right)_{HCO_3^-}}{\left(\frac{^{13}C}{^{12}C}\right)_{OM}} \tag{4}$$

Note that the definition of $\epsilon$ is independent of the isotope reference standard, and thus the unit is given as mUr only. As with delta values, the enrichment factor has to be supplied as a number without units. Internally, ESBMTK only tracks the total concentration and the concentration of the dominant isotope species. The respective delta values are computed once integration has finished.



## 2.2 Weathering

ESBMTK provides a connection type that calculates weathering intensity as a function of $CO_2$. The implementation is rather simple and follows Walker et al. (1981)

$$f = A \times f_0 \times \left( \frac{pCO_2}{p_0CO_2} \right)^c \tag{5}$$

where $A$ denotes the area, $f_0$ the weathering flux at a the reference pressure $p_0CO_2$, The $CO_2$ partial pressure at a given time

$t$, is denoted as $pCO_2$, and $c$ is constant that defines the strength of the weathering (see Walker et al. 1981). It is however easy to add a new weathering class to ESBMTK that adds a more comprehensive parametrization of weathering processes.





## 2.3 Seawater properties and Equilibrium Constants

Provided that the model is specified in units of mol/kg (i.e., substance content, McNaught and Wilkinson 2019), and that pressure, temperature, and the concentrations of total alkalinity (TA) and total dissolved inorganic carbon (DIC) are known, ESBMTK can calculate a variety of tracers and dissociation constants. The carbonate system dissociation constants are taken from pyCO2sys library which unlike hard-coded solutions provides a choice of four different ph-scales, 18 different parametrizations for the dissociation constants, and various methods to calculate buffer factors (see Humphreys et al. 2022). This approach not only avoids code duplication but also simplifies the comparison between different models. At present ESBMTK supports pyCO2sys options to select the pH-scale and the parametrizations for the dissociation constants.

The solubility of $CO_2$ is based on the K0 value returned by pyCO2sys, which follows Weiss (1974). ESBMTK reports the $CO_2$ solubility as `SA_co2` in mol/(t atm) corrected for water vapor pressure at sea level as a function of temperature and salinity following Weiss and Price (1980). The solubility of oxygen follows Sarmiento and Gruber (2006), and seawater density is calculated using the equation of state given by Zeebe and Wolf-Gladrow (2001).

At present, ESBMTK initializes the above parameters at the beginning of each run and assumes that they are constant over the integration interval. In other words, models that require changes in temperature and pressure are currently beyond the scope of ESBMTK.





## 2.4  Carbon Chemistry & Carbonate Dynamics

ESBMTK uses total dissolved inorganic carbon (DIC) and total Alkalinity (TA) as master variables to calculate $[H^+]$ and seawater carbon speciation during integration. The initial $[H^+]$ concentrations in each box are calculated with the pyCO2sys library during the model initialization. The computation of subsequent $[H^+]$ concentrations uses the iterative approach of Follows et al. (2006) where the initial guess of $[H^+]$ is used to calculate carbonate alkalinity, which is then used to calculate a new $[H^+]$. Provided that the changes in $[H^+]$ between integration timesteps are smaller than 3E-11 mol/kg, the associated error is too small to be of concern (Follows et al., 2006). ESBMTK will print a warning if this threshold is exceeded. During integration, ESBMTK only carries tracers for boron and $[H^+]$ and $[CO_2]_{aq}$. All other carbon species are calculated once the integration finishes.

Carbonate dissolution in the water column and sediments is a function of the saturation state which changes with depth. To calculate the resulting burial/dissolution fluxes, one needs a statistical representation of the depth/sediment area relationship in the ocean. ESBMTK approximates this with a hypsometric curve that is based on a 5-minute grid that has been down-sampled from the Global Bathymetry and Topography at 15 Arc Sec (SRTM15+ V2.5.5 dataset, Tozer et al. 2019). The flux calculations use the parametrizations proposed by Boudreau et al. (2010a) and Boudreau et al. (2010b). Their approach first calculates specific depth boundaries (i.e., the saturation depth for $CaCO_3$ $z_{sat}$, or the $CaCO_3$ compensation depth $z_{cc}$) as a function of the average $CaCO_3$ solubility product in the surface ocean ($K^0_{sp}$ = 4.29E-7 mol²/kg²), a characteristic depth value ($z^0_{Sat}$ = 5078), and the calcium and carbonate ion concentrations (see Fig. 2 for equations). In the second step, they provide a parametrization of the resulting $CaCO_3$ burial/dissolution fluxes as a function of the carbonate export flux from the surface ocean and the area between the critical depth intervals (e.g., between $z_{sat}$ and $z_{cc}$). It should be noted that Boudreau et al. (2010a) do not consider the effect of Aragonite dissolution and that their parametrization assumes an idealized mean ocean temperature distribution and homogeneous carbonate ion concentration in the deep ocean (Boudreau et al., 2010b). However, the scheme is computationally efficient and captures transient changes, i.e., times when the snow-line and carbonate compensation depth are at different depth levels.





**Figure 2.** Parametrizations for carbonate burial and dissolution fluxes as proposed by Boudreau et al. (2010a). The letter A denotes cumulative seafloor areas, and the letter B denotes fluxes. The critical depth intervals ($z_0$, $z_{cc}$, $z_{snow}$) denote the separation between the saturated and undersaturated waters, and between carbonate-bearing and carbonate-free sediments. $B_{NS}$ denotes sedimentary calcite dissolution from oxic respiration, $B_{DS}$ denotes the dissolution by respiration in the sediments and in dissolution in the water column, $B_{cc}$ denotes the dissolution below the carbonate dissolution depth, and $B_{PDC}$ the transient dissolution if the depth of $z_{cc}$ and the snowline diverge from each other. $\alpha$ is the fraction of CaCO$_3$ that dissolves above the saturation horizon $z_{sat}$.





## 2.5 Gas Reservoirs and Air-Sea Gas Exchange Fluxes

ESBMTK provides a gas-reservoir class that can be used to track concentration changes of e.g., $pCO_2$. In its default setting, this class uses a mass of 1.78E20 mol for the earth's atmosphere and tracks a given species as the mol ratio relative to the atmosphere. While this class can be used to track several species (e.g., $O_2$ and $pCO_2$), they are currently treated as independent of each other. Further, changes in a given species concentration will not affect the overall mass of the atmosphere. This error associated with typical variations in $pCO_2$ is however negligible.

Gas exchange between two reservoirs is implemented as a connection instance that requires a `GasReservoir` and a regular `Reservoir` instance that carries seawater tracers (see above). The gas exchange implementation follows Zeebe (2012)

$$F_{\text{gas}} = A \cdot u \left( \beta \cdot pCO_2 - [CO_2]_{aq} \right) \tag{6}$$

where $[CO_2]_{aq}$ denotes the concentration of $CO_2$ in solution (in mmol/kg), and $pCO_2$ denotes the atmospheric $CO_2$ concentration (in ppm). $A$ denotes the surface area, $u$ the piston velocity, and $\beta$ the solubility of $CO_2$. Currently, ESBMTK provides these parameters for $CO_2$ and $O_2$.

Isotope fractionation effects related to the exchange of $CO_2$ across the air-sea interface assume that the isotope ratios of $HCO_3^-$ and DIC are roughly equal. This simplification introduces a small error of up to 0.3 mUr at 20 °C and a pH between 7.5 to 8.2 (see Zeebe 2012) and we calculate the gas exchange flux for $^{13}C$ as

$$F_{gas^{13}C} = A \cdot u \cdot \alpha_u \left( \beta \cdot \alpha_{\text{dg}} \cdot p^{13}CO_2 - \alpha_{\text{db}} \cdot R_{\text{T}} \cdot [CO_2]_{aq} \right) \tag{7}$$

where $\alpha_{\text{u}}$ denotes the kinetic fractionation factor during gas exchange (equivalent to and $\epsilon$ value of 0.8 mUr, Zhang et al. 1995), $\alpha_{\text{dg}}$ denotes the equilibrium fractionation factor between $CO_2$ in solution and $CO_2$ in gas ($\epsilon$=1.076 mUr, Zeebe and Wolf-Gladrow 2001), and $\alpha_{\text{db}}$ denotes the equilibrium fractionation between dissolved $CO_2$ and $HCO_3^-$ ($\epsilon$ = 9.36 mUr, Zeebe and Wolf-Gladrow 2001).





## 2.6 Numerical Implementation

ESBMTK defaults to an implicit backward differentiating ODE solver which is suitable for the typically stiff problems in the
earth sciences. Specifically, we use the `scipy.integrate.BDF` solver as provided by the SciPy library which builds on
the algorithms by Byrne and Hindmarsh (1975), Hairer et al. (1993), and Shampine and Reichelt (1997). This algorithm uses
a variable time step and automatically increases the time step until the solution becomes unstable. ESBMTK defaults to an
initial timestep of 1 second. While this seems short given geological time scales, setting this value to a longer time interval has
no perceptible influence on the execution time since the solver rapidly increases the integration interval. Conversely, however,
setting this value too high, can affect the stability of the carbonate system solution. This is particularly true for small-scale
models that, e.g., model the acidification of distilled water in a beaker.

A complication with variable timestep algorithms is however that they cannot know the nature of episodic events, like driving
a model with volcanic or anthropogenic carbon input. The ESBMTK model class thus provides the `max_timestep` keyword
which limits the solver to time step values that are smaller than this value. While the BDF solver is not sensitive to scaling
problems (i.e., differences between variables that are very small and those that are very large), its convergence criterion needs
to be adjusted for variables that differ by orders of magnitudes. ESBMTK does this based on the initial values of the respective
species so that the absolute tolerance value t equals

$$t = 10^{-7} \times v \tag{8}$$

where $v$ denotes a given variable value. In other words, for a concentration value of 28 mM, the solution must be within
±2.8E-6 mM.





## 3   Proof of concept

In order to show the versatility of ESBMTK and to test the model results, we implement the Boudreau et al. (2010a) model
using the ESBMTK library. The model code and associated scripts are available online (see the code availability section below).
The Boudreau et al. 2010a model consists of three ocean boxes, one for the low-latitude ocean areas, one for the high-latitude
ocean areas, and one box for the deep ocean. Additionally, it has a box representing the atmosphere. The model assumes that

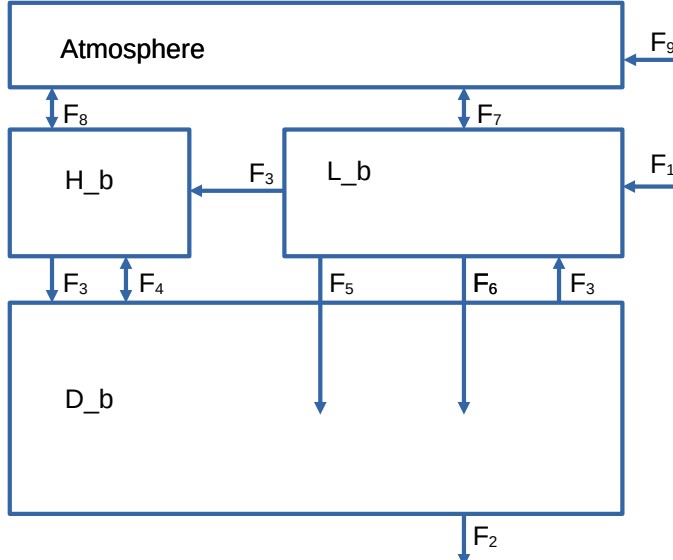

**Figure 3.** Model geometry used by Boudreau et al. (2010b). See text for flux descriptions, and Tab. 2 for flux values. Note that fluxes can
denote more than one species, e.g., F6 stands for the carbonate export flux that will affect dissovced inorganic carbon (DIC) as well as total
alkalinity (TA).

there is no organic and inorganic export flux from the high latitude to the deep ocean box and that the particulate organic matter
flux from the low latitude to the deep ocean box (F5) is fully remineralized and has no effect on alkalinity. The carbonate export
flux (F6) is partly dissolved and partly buried (F2), where the partitioning between F2 and F6 depends on the carbon speciation
in the deep box. The model uses a fixed rain ratio where F5/F6 = 0.3. Alkalinity and dissolved organic carbon are replenished
via a constant weathering flux (F1). The model does not consider phosphor cycling. Thermohaline circulation (F3) and mixing
between the high-latitude and deep ocean boxes (F4) redistribute the dissolved species, and gas exchange with the atmosphere
balances the concentration of dissolved $CO_2$ between the low-latitude and high-latitude boxes (F7 & F8). Model parameters
are given in tables 1 to 4.

Boudreau et al. (2010b) use the equilibrium constants parametrization of Millero et al. (2006), and report their results on
the free pH scale. However, they do not report the fractional value used for the dissolution above the saturation horizon ($\alpha$).
Manual tuning of the ESBMTK implementation suggests that a value of 0.6 results in steady-state conditions that are similar
to the values reported by Boudreau et al. (2010b). We then use these steady-state conditions to force the model with a $CO_2$



| Name | Area [$m^2$] | Volume [$m^3$] | P [bar] | T [°C] |
|---|---|---|---|---|
| $H_b$ | 0.5E14 | 1.76E16 | 17.6 | 2 |
| $L_b$ | 2.85E14 | 2.85E16 | 5 | 21.5 |
| $D_b$ | 3.36E14 | 1.29E18 | 240 | 2 |

**Table 1.** Geometry and PT conditions for the reservoir boxes in the Boudreau et al. (2010a) model. All boxes use a salinity of 35.

| Name | Symbol | Flux |
|---|---|---|
| Weathering DIC | F1 | 12 Tmol/a |
| Weathering Alkalinity | F1 | 24 Tmol/a |
| $CaCO_3$ export | F6 | 60 Tmol/a |
| Organic Matter Export | F5 | 200 Tmol/a |
| Thermohaline circulation | F3 | 25 Sv |
| Mixing | F4 | 30 Sv |

**Table 2.** Flux parameters as used by Boudreau et al. (2010a). The DIC and Alkalinity burial flux F2 is a function of the export productivity and $CO_3^{2-}$ concentration in the deep box (see Fig. 2). The gas exchange fluxes F7 and F8 are a function of the dissolved $CO_2$ concentrations in the surface boxes.

| Parameter | Symbol | Value | Units |
|---|---|---|---|
| Piston velocity | $v_G$ | 4.8 | m/d |
| $CaCO_3$ dissolution coefficient | $k_c$ | 8.84 | m/yr |
| $CaCO_3$ solubility at z=0 | $K_{sp}$ | 4.29E-7 | $mol^2$/kg |
| Characteristic depth | $z^0_{sat}$ | 5078 | m |
| $Ca^{2+}$ concentration | $[Ca^{2+}]$ | 0.0103 | mol/kg |
| $CaCO_3$ inventory | $I_{CaCO_3}$ | 529 | mol/$m^2$ |
| Fraction of $CaCO_3$ dissolution above $z_{sat}$ | $\alpha$ | 0.6 | |

**Table 3.** Biogeochemical rate parameters as used in the ESBMTK version of the Boudreau et al. (2010a) model. With the exception of $\alpha$, all parameters after Boudreau et al. (2010a).





| Box | $L_b$ | $H_b$ | $D_b$ |
|---|---|---|---|
| K0 | 3.1106e-02 | 5.8223e-02 | 5.8223e-02 |
| K1 | 1.0590e-06 | 7.4495e-07 | 9.6431e-07 |
| K2 | 7.5417e-10 | 4.1328e-10 | 4.9063e-10 |
| KW | 3.5299e-14 | 5.6521e-15 | 6.9213e-15 |
| KB | 1.8545e-09 | 1.2038e-09 | 1.6189e-09 |
| DIC [$\mu$mol/kg] | 1940 (1952) | 2151 (2153) | 2294 (2291) |
| TA [$\mu$mol/kg] | 2281 (2288) | 2347 (2345) | 2403 (2399 |

**Table 4.** Equilibrium constants $K_i$ as used in each box. These values are computed by pyCO2sys (Humphreys et al., 2022) based on the PT values in Tab. 1, and reported relative to the free pH scale. The concentrations values for DIC and TA are the steady-state concentrations in the ESBMTK version of Boudreau et al. (2010a). The steady-state values of the original model are in brackets. The steady-state $pCO_2$ in the ESBMTK model is 275 ppm, Boudreau et al. (2010a) do not list their steady-state $pCO_2$.

pulse (F9) that is based on the IS92a emission scenario (Leggett et al., 1992) but uses a Gaussian evolution after 2100 AD that peaks near the year 2250 AD. The total $CO_2$ emission equals 4025 Gt C over 600 years, and Boudreau et al. (2010a) assume that there is no terrestrial carbon uptake. Fig. 4 shows a comparison between the ESBMTK-based model implementation and the data reported by Boudreau et al. (2010a).

Both models demonstrate that the $CO_2$ release (Fig. 4 panel g) increase the $CO_2$ fluxes across the air-sea interface (panel 185  d) and the increase in ocean water acidity due to the dissolution of $CO_2$. This causes a rapid rise of the saturation horizon (zsat, panel e), a fairly rapid rise of the carbonate compensation depth (zcc, panel e), and a slower rise of the snow line (znow, panel e). Consequently, the carbonate burial flux decreases, and the carbonate dissolution flux increases (panel h), elevating the DIC and TA concentrations in all ocean boxes. The increase in TA, enhances the oceans buffer capacity, leading to a rapid drawdown of atmospheric $CO_2$ after the year 2320 (panel f). However, returning to preindustrial steady-state values requires 190  the re-equilibration of the marine alkalinity pool, a process that occurs over hundreds of thousands of years. For a detailed interpretation of the model results refer to the original publication by Boudreau et al. (2010a).

## 4  Discussion

The steady-state results of the ESBMTK model broadly match the data of Boudreau et al. (2010a), but also show noticeable differences. This is particularly true for the low-latitude ocean where both the DIC and TA steady-state concentrations are 195  lower than those in the original model (12 and 7 $\mu$mol respectively, see Tab. 4) which in turn affects the gas exchange fluxes (panel d in Fig. 4). In the deep box, the DIC concentration is 2.6 $\mu$mol higher and the TA concentration is 4 $\mu$mol higher than in the original model, resulting in a slightly higher $CO_3^{2-}$ concentration (87 versus 86 $\mu$mol/kg in the original model), deepening the location of the critical horizons by about 50 meters.





**Figure 4.** Comparison between the models results reported by Boudreau et al. (2010a) and the ESBMTK-based implementation. Solid lines denote the ESBMTK results, dotted lines denote data that has been digitized from the figures in Boudreau et al. (2010a). See text for discussion.





The differences between the low latitude surface box and the deep ocean are mainly controlled by the export productivity and
the burial/dissolution fluxes as well as the thermohaline upwelling. Productivity and upwelling velocity are known constants,
and the burial dissolution fluxes equations are known as well. However, the fraction of carbonate dissolution ($\alpha$) above the
saturation horizon is not mentioned by Boudreau et al. (2010a). Increasing $\alpha$ until the surface DIC and TA values are a
better match with the original model, increases however the differences in the deep box. This in turn increases the $CO_3^{2-}$
concentrations and deepens the depth of the $z_{Sat.}$, $z_{cc}$ and $z_{snow}$ horizons by another 50 meters, and further reduces the steady-
state $pCO_2$. Carbon speciation in the deep box would also be affected by the choice of dissociation constants, but it is also
conceivable that the differences are caused by underlying hypsographic data.

We cannot exclude the possibility that there is a numerical error in the ESBMTK library, but it is more likely that the
observed variations are caused by small differences in the dissociation constants, and or hypsometric data. Both, ESBMTK
and Boudreau et al. (2010a) use the carbon dissociation constants parametrization of Millero et al. (2006), however, both rely
on third-party libraries (pyCO2sys and AquaEnv, respectively) to calculate the k-values and we were unable to compare the
constants used in our model with the constants used in the original model.

Boudreau et al. (2010a) provide a non-steady state case to test the response of the system against the release of 4025 Gt over
600 years. We digitized the forcing function for our model from Fig. 2 in Boudreau et al. (2010a). Integration of the digitized
data yields a total carbon mass of 4590 Gt C instead of 4025 Gt. We, therefore, scale the digitized data by a factor of 0.877,
which results in the differences shown in panel g) of Fig. 4. Using the solid line in panel g) as a forcing function, our model
yields results that are similar to the original model. While the $CaCO_3$ burial and dissolution fluxes are similar, the long-term
response in the deep ocean alkalinity is among the more visible differences. However, overall, the ESBMTK implementation
replicates the result of the original model well.

## 5 Conclusions

ESBMTK started as a teaching tool, with the idea to emphasize model geometry and processes over coding details. This is
particularly true for conceptually simple models in combination with Jupyter Notebooks, an approach that has been successfully
used in undergraduate classes that had no previous coding experience. Advanced students with basic Python skills benefit from
using ESBMTK by being able to focus on the inherent complexities of model definition, rather than being sidetracked by
numerical issues. This approach significantly reduces model development time and ensures that the object-based modeling
results in well-documented code that is easy to read with a basic understanding of Python syntax. The hierarchical, object-
oriented program structure provides a robust framework for experienced Python programmers to adapt or extend the ESBMTK
library. These features are also attractive in a research environment, significantly improving readability and reproducibility
without incurring major performance penalties.

Rather than implementing our parametrizations for the various equilibrium constants, we use the well-tested pyCO2sys
library which provides access to a wide range of published equilibrium constants and a choice of different pH scales. At present,
carbonate chemistry computations are based on previously published algorithms that are suitable for the modern ocean, but



will require adaptions for conditions where the deep ocean is warmer than today. Likewise, at present the model is only valid for modern ocean Ca and Mg concentrations and only considers calcium carbonate, but not aragonite. Re-implementing a previously published model that uses the same carbon chemistry algorithms, we find that the results of both models are in good agreement. We do observe however small differences which we attribute to minor variations in the underlying carbon species equilibrium constants.

## 6 Code availability

The current ESBMTK version is available through the conda and pip package managers and from the project website: https://github.com/uliw/esbmtk under the GNU Lesser General Public License v3.0. The documentation is available at https://esbmtk.readthedocs.io and example scripts including the model described in this paper are available at https://github.com/uliw/ESBMTK-Examples. The exact version of the model used to produce the results used in this paper is archived on Zenodo (https://doi.org/10.5281/zenodo.11959366), as are input data and scripts to run the model and produce the plots for all the simulations presented in this paper (https://zenodo.org/doi/10.5281/zenodo.12100595).

## 7 Author Contribution Statement

UGW initiated this project, secured funding and wrote the manuscript. TT and MN implemented the carbonate chemistry module and used ESBMTK to implement the Boudreau 2010 model. RN wrote the first version of the ODE backend, JZ and MW developed the hypsometry code, BSC and MRM the phosphorous feedback module.

## 8 Competing Interests

The authors declare that they have no conflict of interest.

## 9 Acknowledgments

UGW was supported through an NSERC Discovery Grant and a fellowship at the Hanse Institute for Advanced Studies. TT was supported through an NSERC USRA award, MRM through a CGCS internship, JZ through a UofT UTEA award, BSC CAPES PDSE scholarship award (Process: CAPES-PRINT - 88887. 935155/2024-00). Jörg Bollmann provided helpful comments on an early draft of thus manuscript.



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
