# Peer review of "The Earth Science Box Modeling ToolKit (ESBMTK 0.14.0.11): A Python Library for Research and Teaching"

_EGUsphere, 2024_

## Author Response (AR1)

**Point by point reply to the reviewers comments**

Note: Review comments in italics.

**Review #1**

*Wortmann et al. present the Earth Science Box Modeling Toolkit (ES-BMTK), which is a Python library designed for building and analyzing box models in Earth science. It uses a modular, object-oriented approach to study topics like the long-term carbon cycle and the impact of atmospheric CO2 changes on seawater chemistry. ESBMTK separates model geometry from the underlying numerical implementation, and thus allows users to focus on the conceptual challenges, rather than mathematical theory. Such a tool is very useful for teaching and research requiring fast conceptual models. In addition to predefined setup, the user can customize rather easily his/her own model such as the number and volume of boxes/reservoirs, the flux between them, the isotope species, . . . This tool will be very useful for the climate community, and I therefore recommend publication after minor revisions.*

We would like to thank the anonymous referee for their time and thoughtful comments. Below is our detailed response:

1. *Several species are already defined in species$_{definition.py}$ such as stable water isotope 2H and 18O, 13C, . . . I would recommend to add the 14C because it is quite usual to use ocean box modeling to model 14C, especially in the framework of IntCal. See for example Bard et al. (1997, $https://doi.org/10.1016/S0012-821X(97)00082-4$).*

   - The idea to add $^{14}$C as species is a good one, and we can see the utility of it, even so it is outside our own expertise. We added $^{14}$C & $\Delta^{14}$C to the list of carbon species, however, that is different from a full implementation of how $^{14}$C fractionates relative to $^{12}$C during air-sea transfer, photosynthesis, and CO$_2$ speciation. This would require that ESBMTK had the necessary data structures to express isotope systems with more than 2 components, as well as the necessary code to deal with radioactive decay. Given the universal nature of the library (as opposed to a specific model), this is no easy feat to implement, and beyond the scope of the current manuscript. We are however grateful for the suggestion as this is something we have not thought about, but

will keep in mind for future revisions, and we added the following sentence at line 61 to clarify this limitation. It reads:

– Several ESBMTK classes have the option to perform stable-isotope-related calculations, with the important caveat that presently there is no structure for isotope systems with more than two, or radiogenic isotopes.

2. *In my opinion, it would be beneficial to give more details on the modeling of stable water isotopes 2H and 18O, and not only on the carbon-related species. The transport between the ocean box models are quite simple in that case (one to one), but the authors could mention how to set up a simulation when considering fractionation effect between atmosphere and ocean boxes e.g, the evaporation from the ocean to the atmosphere.*

   • We revised the isotope chapter of the manual and now show an example how to setup oxygen isotope exchange reactions during air-sea transfer, using the preconfigured values for oxygen. While we do not have the relevant isotope fractionation factors for hydrogen at hand we now additionally provide an example how to set setup oxygen isotope exchange reactions with user-defined values for the exchange and fractionation coefficients. We now write in line 153:

      – ESBMTK provides the respective fractionation factors for $CO_2$ and $O_2$ . For other gases, these factors can be specified in the connection properties (see Listing 4)

3. *The authors discuss the ESBMTK results with Boudreau et al. (2010a) model. They show the good ability of ESBMTK to replicate other models, which is important to encourage potential users to switch to ESBMTK. One other important aspect is to show how realistic model results are. Is there any way for the authors to compare the ESBMTK results with observations for a typical simulation (or with the setup of Boudreau et al. (2010a))?*

   • We are not sure how to respond to this comment. The MS already provides a detailed comparison with the Boudreau model. A comparison of the Boudreau model with observational data is outside of the objective for this paper, and would involve a detailed discussion on how to map the GLODAP data into an 3-box ocean model.

**Review #2**

*This manuscript introduces the Earth Science Box Modeling Toolkit (ES-BMTK), a Python library designed to streamline the build and operation of box models. The toolkit incorporates many commonly used processes such as air-sea exchange, marine carbonate chemistry, and isotope calculations. Its feasibility and robustness are validated through replicating the work by Boudreau et al., 2010.*

*Given the significant utility of box models in the Earth Science, ESBMTK has considerable potential for applications in both teaching and research settings. Below are several suggestions that, hopefully, could help further enhance the quality and impact of this project:*

We would like to thank Shihan Li for their time and thoughtful comments. Below is our response to the specific points raised by this review:

1. *The toolkit's ability to separate model geometry from its underlying numerical implementation is highly convenient, particularly for users without extensive programming experience. However, this abstraction also hides the core functions, making debugging more challenging. One potential enhancement could be enabling the model to output the codes for governing equation it generates during runtime. Such a feature would not only improve the model transparency but also serve as a valuable resource for educational purposes, allowing users to better understand the mechanics of the model. .... 2) /Several processes and their parametrizations are described in Section 2. However, I found it somewhat challenging to clearly link these processes to the toolkit framework as depicted in Figure 1. It would be helpful if the authors could integrate these processes explicitly within the framework illustration or provide some example codes.*

   - This is a good point. We revised Fig. 1 and its caption, added a short example code for isotope fractionation and weathering (see listing #2 & #3, page 6 & 7) , and refer the reader to these listings where apropriate. E.g., line 80 now reads:
     - Adding, e.g., isotope fractionation to a given connection (transport process), requires that the respective reservoirs have been initialized with a defined isotope ratio, and that the connection instance specifies the fractionation factor (see Listing 2).

3. *I believe perturbation experiments are crucial for exploring system behavior under various conditions. I was happy to learn from the documentation that the model accepts external forcings. I suggest highlighting this feature more prominently and giving brief introduction on its implementation in the main text*

   - We are thankful for the suggestion and added an additional paragraph to the MS as well as some example code (Listing #5). Lines 155 ff read:

     – A key element in box modeling studies is to force one or more model boundary conditions, e.g., $CO_2$ emissions. ESBMTK provides the Signal class that implements methods to create square, pyramidal, and bell shaped signals, as well a method to read forcing data from a CSV-file. The signal data can either be interpreted as an absolute flux that is added to an existing flux, or as a multiplier that is used to increase/decrease a given flux. Furthermore signal instances can be added together to create arbitrarily complex shapes. Signal data is automatically truncated and/or padded to match the model time domain, and the data is resampled so that it matches the model time step. However, care must be taken that signal duration is at least four times as long as the model time step. Signal instances are then associated with one or more connection instances. See the code example in listing 5.

4. *Several things might need further clarifications. a & b) Line 99, the authors state that "the above parameters at the beginning of each run and assumes that they are constant over the integration interval." However, it is unclear what "the above parameters" specifically refer to. My interpretation is that this might relate to thermodynamic and kinetic constants. However, the carbonate system is typically time-dependent and calculated in each integration. . . . Again, from the same paragraph, the authors state that modeling temperatures remain constant throughout the runtime. However, this seems inconsistent with examples where transient and long-term carbon cycles are simulated. Since temperature can affect thermodynamic constants in the carbon system, this assumption may not be realistic. If my understanding is incorrect, additional explanation would be appreciated to resolve this confusion.*

   - Yes, currently the thermodynamic and kinetic constants are calculated only once at the beginning of the run, and then held constant. We have done this since in many cases (e.g., the glacial-interglacial ocean) the changes to the to thermodynamic constants of the carbonate system are for all practical purposes fully compensated by the alkalinity changes that result from ocean volume changes (Zeebe and Wolf-Gladrow, 2001). It is thus a common practice for many of the published box-models, since the calculation of the thermodynamic constants is computationally expensive. We are aware however that in models where ocean temperature changes faster than ocean volume (e.g., as a result of anthropogenic carbon release) the above can be a limitation. We therefore are actively working to implement a feature that will recalculate the thermodynamic constants if pressure/temperature change during the model run. However, this will have to wait for a future version of the library. To address the current concern, we have revised the working in the MS and now clearly state the current approach, its rationale and its limitations. We state on line 104:

– It should be noted that presently, the code assumes that neither temperature nor pressure change during the model run. Therefore thermodynamic and kinetic constants are not updated during the model run. In many cases this is of no concern since, e.g., during glacial-interglacial changes, the changes to the carbonate equilibrium constants are almost fully compensated by the change in ocean volume and the resulting variations in alkalinity (Zeebe and Wolf-Gladrow, 2001). However, this is not universally true and remains an important tradeoff between computational efficiency and precision. Future releases will alleviate this shortcoming.

– And on line 253 in the Conclusions: "We also note that the current 0.14.x version of the library does not update kinetic and thermodynamic constants during model execution"

5. *Line 105, It is noted that [H+] is initially calculated using thepyco2sys library, and subsequently, the iterative approach of Follows et al. (2006) is applied. I'm confused by the rationale behind using two different methods for this calculation.*

• The carbonate system is fully defined if we know at least two of it components. If we, e.g., know the concentration of total alkalinity (TA) and dissolved organic carbon (DIC), we can calculate all other carbon species. This is however computationally expensive. Follows et al. (2006) showed that if we have a reasonable guess for the initial $[H^+]$ at time $i-1$, one can directly compute the concentration of all carbon species with sufficient precision based on the concentrations of TA and DIC at time $i$. This is computationally highly efficient, but relies on a suitable initial estimate for $[H^+]$ which we obtain from pyco2sys. We revised the wording the manuscript so that our rationale is clearly stated. Lines 110ff now state:

- While TA and DIC fully determine the state of the marine carbonate system, solving for $[H^+]$ is computationally expensive. Follows et al. (2006) demonstrate that if one knows a suitably close estimate for $[H^+]$ at t=i, one can estimate $[H^+]$ at t=i+1 with sufficient precision from the concentrations of [DIC] and [TA] without computational overhead. Provided that the changes in $[H^+]$ between integration time steps is smaller than 3E-11 mol/kg, the associated error is too small to be of concern (Follows et al., 2006). We therefore use the pyCO2sys library during the model initialization to compute an initial initial $[H^+]$ concentration, and the use iterative algorithm of Follows et al. (2006) in subsequent time steps.

6. *Carbon isotopes are crucial tracers in carbon cycle modeling. In Example 4 (`https://github.com/uliw/ESBMTK-Examples/blob/main/Examples_-from_the_manual/po4_4.png`), the steady-state surface box $^{13}C$ reaches approximately 8‰, which is not realistic. Since the Boudreau et al. (2010) model does not incorporate isotope modeling, I recommend that the authors conduct additional validation work for this component.*

- The carbon-isotope example in the manual was never meant to be realistic (for that matter neither is the P-cycle model), it was merely a demonstration on how to do it. Achieving realistic DIC isotope values would require that we also add carbonate precipitation and dissolution which would result in a rather complex model. To address this concern, we revised the manual based on the suggestions of reviewer #1 and now show simple examples where the outcome of the isotope fractionation process is predictable (i.e., mixing of two reservoirs, equilibrium fractionation, etc.) We would like to note that the isotope calculations

are covered by unit-tests and that internal testing also involves an unreleased ESBMTK implementation of the LOSCAR isotope model (Zeebe, 2012) which allows us to compare the carbon isotope calculations against published results.

**References**

Follows M.J., Ito T. and Dutkiewicz S. (2006) On the Solution of the Carbonate Chemistry System in Ocean Biogeochemistry Models. *Ocean Modelling* **12(3-4)**, 290–301. doi:10.1016/j.ocemod.2005.05.004.

Zeebe R.E. (2012) LOSCAR: Long-term Ocean-atmosphere-Sediment CArbon cycle Reservoir Model v2.0.4. *Geoscientific Model Development* **5(1)**, 149–166. doi:10.5194/gmd-5-149-2012.

Zeebe R.E. and Wolf-Gladrow D.A. (2001) *CO$_2$ in Seawater: Equilibrium, Kinetics, Isotopes.* Number 65 in Oceanography Book Series, Elsevier.